# RAPTOR: Reasoned Agentic Portfolio Trading with Orchestrated Rebalancing

## Abstract

We propose an institutional-style, multi-agent architecture for equity portfolio construction that couples a schema-constrained blackboard with structured debate and a Black–Litterman optimizer, enabling diversified, risk-aware allocations. The system is scalable, modular, and interpretable, allowing agents to asynchronously exchange structured messages, negotiate tradeoffs, and dynamically rebalance portfolios. We analyze the performance of our approach in a biweekly, one-year reconstruction study on S&P 500 constituents, demonstrating that this combination of structured agent collaboration and Bayesian portfolio blending is practical and modular. We also operationalize a Chain-of-Alpha methodology, and allow classical indicators, for example, Moving Average Convergence Divergence (MACD) and Relative Strength Index (RSI) to serve as checkable features within each chain. Our approach achieves a return of 13.43% over the 8 month backtest, exceeding the S&P 500's 10.08% over the same time frame. The source code and data sets used are available anonymously at https://anonymous.4open.science/r/RAPTOR-Reasoned-Agentic-Portfolio-Trading-with-Orchestrated-Rebalancing

## 1 Introduction

Traditionally, institutional hedge funds coordinate specialist teams to construct, adjust, and hedge portfolios. Recent Large-Language Model (LLM) frameworks emulate this with multi-agent systems, but most rely on sequential free-text exchanges without persistent structured memory, making uncertainty hard to quantify and limiting scalability to portfolio-level hedging and optimization Xiao et al. [2024], Luo et al. [2002].

We propose a modular multi-agent architecture that mirrors institutional practice to answer our research question "When LLM agent views are integrated via Black–Litterman (BL), do portfolios achieve better risk-adjusted performance than (i) market beta (SPY) and (ii) equal-weight portfolios?" A schema-constrained blackboard serves as shared memory: agents write/read JavaScript Object Notation (JSON) messages to an append-only log, yielding auditable, queryable context and avoiding the "telephone game" of unstructured chat. Agents engage in formal cross-examination (bull vs. bear researchers; risk managers with conservative/neutral/aggressive profiles). Their confidence-weighted views are fused with a Black–Litterman mean–variance optimizer, combining equilibrium priors with agent beliefs to produce diversified, risk-aware allocations. We also implement a Chain-of-Alpha (volatility, directional, hedging) with classical indicators (e.g., MACD, RSI) as checkable features.

We evaluated a year of multimodal real-world data across the S&P 500 constituents with biweekly rebalancing. The system supports asynchronous agent collaboration, transparent rationale tracing, and scalable portfolio construction that current LLM-based approaches lack.

Submitted to 39th Conference on Neural Information Processing Systems (NeurIPS 2025). Do not distribute.

## 2   Related Works

Early multi-agent trading systems such as MASST Luo et al. [2002] and PROFTS Reis [2019] used role-based agents coordinated via blackboard-style architectures, combining specialized signals (e.g., technical, news, valuation). Despite promise, these approaches were largely rule-based and inflexible, with limited probabilistic reasoning.

The rise of large-language models has transformed multi-agent system (MAS) frameworks. TradingAgents Xiao et al. [2024] simulates institutional workflows specialized agents via natural language but it can suffer from verbosity and context loss. FinCon Yu et al. [2024] adds hierarchical analyst–manager coordination and belief refinement, but remains restricted to small equity sets and informal communications. HedgeAgents Li et al. [2025] introduces asset class-specific hedging agents and structured "conferences", but lacks persistent structured memory and broader portfolio optimization.

More recently, works that fuse LLM outputs with portfolio theory, such as integrating LLM-generated views into a Bayesian Black–Litterman mean–variance framework Lee et al. [2025] or role-based equity construction with AlphaAgents Zhao et al. [2025], report robustness gains and surface strengths/limits across risk tolerances.

Structured communication and memory remain key fields of inquiry. Outside of financial trading, systems like MetaGPT Hong et al. [2024] and AgentVerse Chen et al. [2023] demonstrate the benefit of structured protocols and shared memory to mitigate information decay; TradingAgents partially adopts structured reports, but still lacks persistent, queryable, schema-constrained memory.

## 3   Method

Our system follows an institutional-style pipeline that combines structured multi-agent collaboration with Bayesian portfolio optimization. First, each asset in the investment universe is assigned to a thread that coordinates multiple specialized agents, consisting of analysts, researchers, and risk managers, who communicate exclusively through a schema-constrained blackboard using typed JSON messages. These agents collect data, debate bullish and bearish theses, and generate categorical BUY/HOLD/SELL views along with confidence indicators. The resulting views are then aggregated and translated into numerical inputs for a Black–Litterman optimizer, which blends equilibrium priors with agent-derived expectations to produce risk-aware portfolio weights. The final allocations are rebalanced on a fixed cadence and evaluated over time to measure returns, volatility, and risk-adjusted performance. This modular structure ensures transparency, scalability, and traceability from agent rationales to portfolio outputs.

### 3.1   Blackboard Communication Protocol

The blackboard system implements a centralized communication architecture in which agents post structured messages to a shared repository and read relevant information from other agents. This system enables asynchronous, decoupled communication between agents through typed message schemas (AnalysisReport, TradeProposal, RiskAlert, etc.), allowing for coordinated decision-making while maintaining agent autonomy. The blackboard facilitates information sharing, debate, and collaborative analysis throughout the multi-agent trading framework while allowing human interpretability through viewing each agent's contribution to the decision-making process. Agents write messages in an append-only JSONL log, with each entry conforming to a typed schema (e.g., AnalysisReport or TradeProposal) containing fields such as sender, intent, timestamp, ticker, and structured content. Retrieval is done via lightweight filtering on attributes like message type, sender role, ticker symbol, or recency, so each agent only reads the most relevant recent messages rather than scanning the entire log.

### 3.2   Agent Roles

The system employs four groups of agents: **analysts** (fundamental, macro, market, news, social) collect data; **researchers** (bull/bear with cross-exam) debate and, under a research manager, synthesize theses into final reports; **risk managers** (conservative, neutral, aggressive) follow a similar

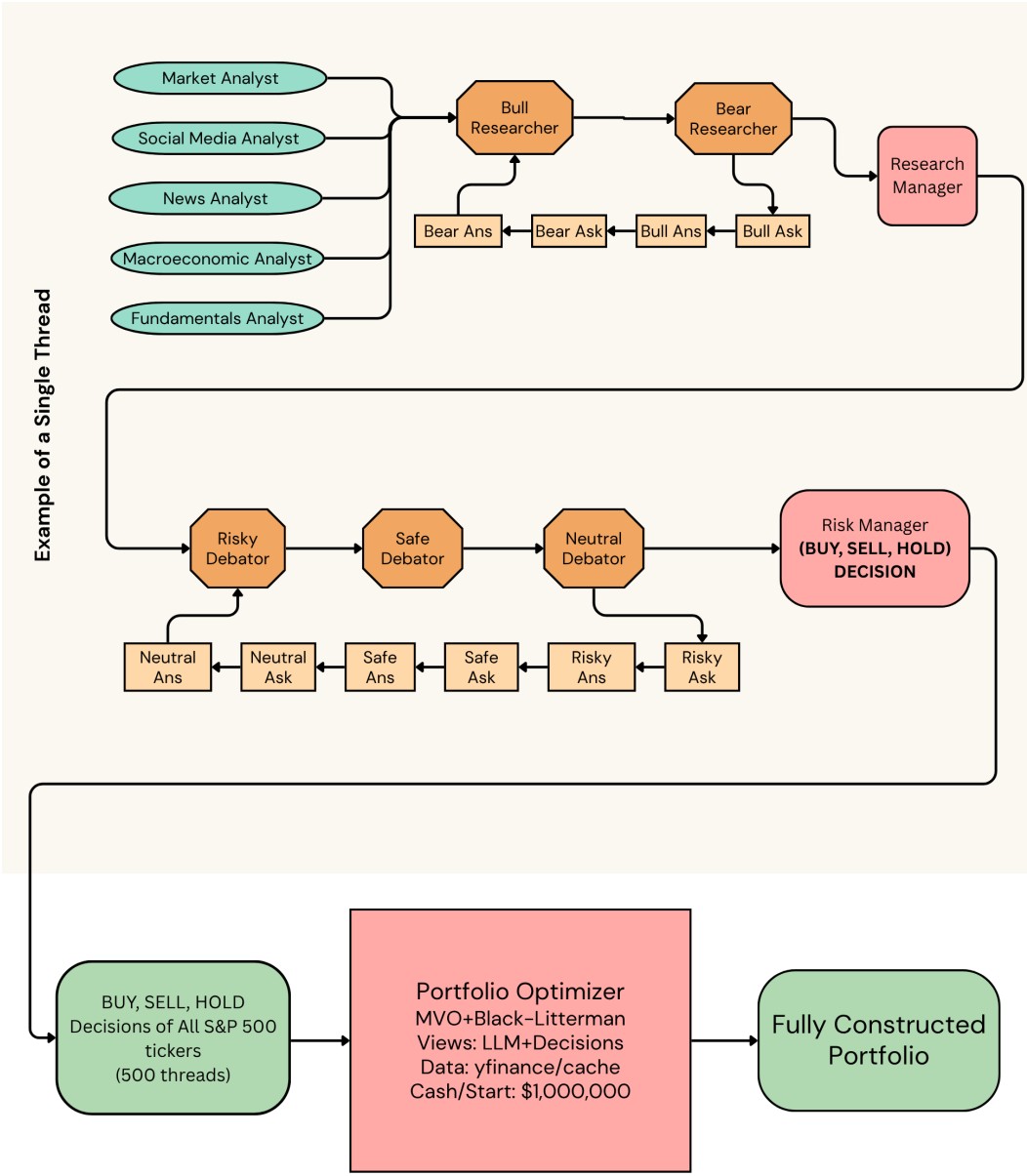

Figure 1: End-to-end pipeline from per-ticker analysis (BUY/SELL/HOLD) to Black–Litterman portfolio optimization.

ask/answer protocol, with a judge consolidating perspectives into risk assessments; and **execution agents** (trader, optimizer) translate outputs into allocations via Mean-Variance Optimization (MVO)/Black–Litterman (see Fig. 1). Both researcher and risk debates are coordinated by conditional logic that controls sequencing, round limits, and state transitions.

At the end of the researcher debate, the research manager compiles the strongest bull and bear arguments into a structured FinalReport message containing the ticker, thesis summary, supporting evidence, and an aggregate BUY/HOLD/SELL view with confidence. This report is then posted to the blackboard and automatically retrieved by the risk managers, who open a similarly structured debate focused on exposure, drawdown tolerance, and scenario stress testing. Debate termination is controlled by predefined logic that ends the exchange after a fixed number of turns (typically 2–3 rounds per side), or earlier if consensus or no new evidence emerges. The risk judge then produces a RiskAssessment message in the same JSON schema format, which becomes the final input handed to the execution agent and Black–Litterman layer.

## 3.3 Incorporation of LLM-Generated Views in Black–Litterman

$$\mu = \left(\tfrac{1}{\tau}\hat{\Sigma}^{-1} + P^\top \Omega^{-1} P\right)^{-1}\left(\tfrac{1}{\tau}\hat{\Sigma}^{-1}\pi + P^\top \Omega^{-1} q\right), \tag{1}$$

$$\min_{w} \quad \frac{\lambda}{2} w^\top \hat{\Sigma}\, w \; - \; \mu^\top w \tag{2}$$

$$\text{s.t.} \quad \mathbf{1}^\top w = 1, \tag{3}$$

$$w \geq 0 \quad \text{(optional long-only)}, \tag{4}$$

$$\ell \leq w \leq u \quad \text{(optional bounds)}.$$

We then construct the posterior in the usual way. Each asset receives a view $q_i \in \{+2\%, 0\%, -2\%\}$ (annualized) according to the agent output. The confidence matrix $\Omega$ is diagonal with entries

$$\Omega_{ii} = 0.5\,\tau \left[\text{diag}\big(P\,\widehat{\Sigma}\,P^\top\big)\right]_i,$$

where the confidence scaling factor is $0.5$. The selection matrix is $P = I_n$, so all retained tickers are treated as absolute views. We estimate the prior covariance $\widehat{\Sigma}$ using the sample covariance on a 252-trading-day lookback (annualized), and derive the equilibrium prior $\pi = \delta\,\widehat{\Sigma}\,w_{\text{mkt}}$ with risk-aversion $\delta = 3.0$. We blend prior and views via Eq. (1) using $\tau = 0.025$. In the unconstrained case, the optimized weights are given by

$$w^\star = \frac{1}{\lambda}\Sigma_{\text{BL}}^{-1}\,\mu_{\text{BL}},$$

then normalized so that $\mathbf{1}^\top w^\star = 1$. Optional long-only constraints are imposed by element-wise clipping of negative components.

We adopt the standard Black–Litterman formulation but integrate LLM-generated views from the multi-agent pipeline. Each included ticker receives an absolute view under the identity selection matrix $P = I$, where BUY, HOLD, and SELL decisions map to $+0.02$, $0.0$, and $-0.02$, respectively. These $\pm 2\%$ annualized returns serve as conservative directional adjustments that do not overpower the equilibrium prior and function as a proof-of-concept baseline rather than tuned parameters. We use a diagonal confidence matrix $\Omega$ because agent debates and view formation occur independently per ticker, and we do not currently estimate cross-asset correlations in LLM-derived signals. This choice also preserves computational tractability and aligns with common BL implementations under view independence. Future work may replace these fixed magnitudes and diagonal assumptions with confidence-weighted or correlation-aware views derived from agent-level consensus or historical signal performance.

# 4 Results and Experimentation

We begin by outlining the datasets, timing hygiene, and evaluation protocol used for our daily, point-in-time backtests on S&P 500 constituents. We then present core performance and risk metrics, followed by ablations and interpretability analyses linking agent debates to Black–Litterman allocations.

## 4.1 Data

For the experimental setup, we collected data from **FinnHub**, **YFinance**, **Reddit**, **SimFin**, and **Perplexity** to support portfolio construction, optimization, and evaluation. Specifically:

- **FinnHub**: Company news, insider sentiment/transactions, and SEC filings (including quarterly reports). Queried dynamically by the News and Fundamentals analysts using the trading date as the endpoint with short lookback windows (no fixed "2025-01-01" snapshot).

- **YFinance**: Historical daily equity prices available from 2015–01–01 through 2025–07–27 for S&P 500 constituents. Used throughout with rolling 252-day lookbacks to compute returns and covariance estimates.

- **Reddit**: r/wallstreetbets posts collected over the range 2025–01–01 to 2025–08–19; at inference time, posts are filtered per trading date using a 7-day rolling window by the Social Media analyst.
- **SimFin**: Quarterly financial statements (income statement, balance sheet, cash flow) consumed by the Fundamentals analyst to inform LLM-driven views; SimFin signals affect Black–Litterman indirectly via the agents' categorical decisions (they are not passed to BL as raw fundamentals).
- **Perplexity**: Macroeconomic news artifacts loaded as JSON per trading date and used by the Macroeconomic analyst to contextualize agent reasoning.

All sources are accessed dynamically per trading date when the full multi-agent pipelines are executed. In the multithreaded backtesting script, comprehensive agent pipelines run primarily on the first backtest day ("Day 0"), after which subsequent dates use MVO/BL rebalancing driven by the previously generated decisions; tickers lacking sufficient trailing price history are filtered out prior to optimization, and no cross-thread data sharing occurs during analysis (synchronization happens only at decision aggregation).

## 4.2 Experimental Setup

We simulate a trading horizon from 2025–01–01 to 2025–08–29. On the first trading date ("Day 0"), we execute the full multi-agent pipeline in parallel across eligible tickers, using one thread per asset. Each thread runs analysts, researcher debates, and risk assessments end-to-end, writing blackboard messages to a shared global JSONL log (isolation achieved via ticker-based message filtering) and persisting longer-term memory to an isolated ChromaDB collection identified by a unique memory suffix. Once all threads complete, the resulting BUY/HOLD/SELL decisions are aggregated and passed to the Black–Litterman (BL) optimization stage.

On each trading date in our evaluation, we rerun the complete multi-agent pipeline in parallel across eligible tickers (analyst reports, bull/bear researcher cross-examination, and risk assessment), writing messages to the blackboard for that day and producing fresh BUY/HOLD/SELL decisions. We then compute that date's allocations and log outcomes. We do not cache Day 0 views or apply a fixed rebalancing cadence; evaluation proceeds at a daily frequency.

Execution is parallelized with a thread pool. There is no cross-thread state or memory sharing during analysis (each thread uses a distinct ChromaDB collection); synchronization occurs only when final per-ticker decisions are collected prior to portfolio optimization. BL portfolio weights are computed at each rebalance with optional long-only constraints.

## 4.3 Results

Our strategy outperformed the S&P 500 benchmark during the eight-month backtest. The portfolio achieved a return of 13.43% over the eight month testing period, compared to 10.08% for the benchmark (Fig. 2), yielding an excess of 3.35 percentage points. This indicates that the framework was able to generate nontrivial alpha rather than merely tracking market beta.

Risk-adjusted performance was evaluated using a 20-day rolling Sharpe Ratio. The overall Sharpe reached 1.0, substantially above typical market baselines, indicating strong efficiency in the realized risk–return profile. However, the ratio fluctuated widely over the horizon (range: $-2.42$ to $5.27$ with a mean of $1.41$ and a standard deviation of $2.63$ month–to–month), reflecting both sharp drawdowns and rapid recoveries. These swings highlight the strategy's sensitivity to market regimes and the potential for instability, despite robust overall performance.

**Extended Validation and Robustness Analysis.** To further substantiate the performance of the system, we performed a comprehensive validation using daily portfolio data from 2025-01-01 to 2025-08-29 ($N = 166$ trading days). The portfolio's value increased from \$1,000,000.00 to \$1,134,348.19, yielding a total return of 13.43% and an annualized return of 21.09%. The portfolio exhibited an annualized volatility of 19.30%, a Sharpe ratio of 1.0 (risk-free rate $r_f = 2.00\%$), a Sortino ratio of 1.28, and a maximum drawdown of –15.33%.

Rolling Sharpe analysis revealed that short-horizon windows capture transient volatility effects (mean 20-day Sharpe $1.60 \pm 3.28$), while longer windows stabilize around $1.1-1.4$. This confirms that the

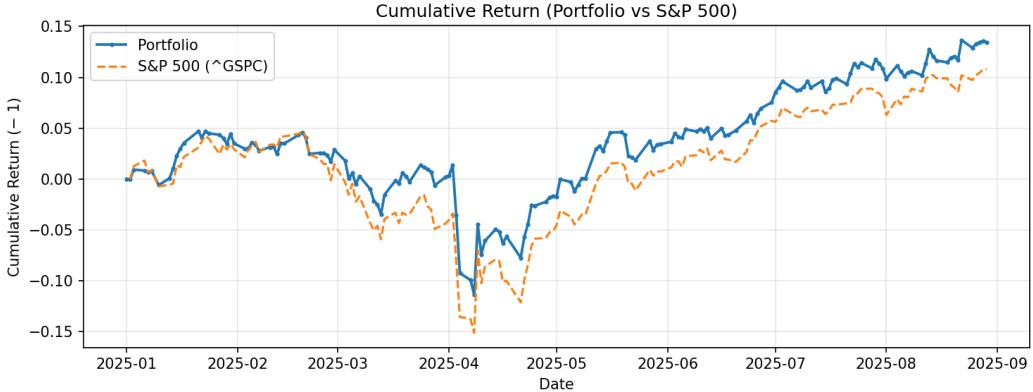

Figure 2: Cumulative returns of the optimized portfolio compared to the S&P 500 benchmark over the backtest period from January 1, 2025, to August 29, 2025. The portfolio outperformed the benchmark, demonstrating the effectiveness of the optimization strategy.

full-period Sharpe of approximately $1.0$ accurately reflects steady risk-adjusted efficiency over the eight-month evaluation.

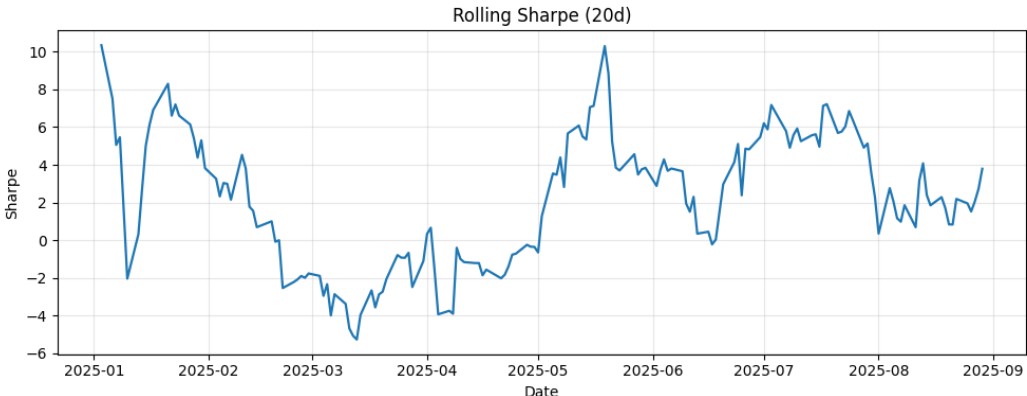

Figure 3: Rolling 20-day Sharpe Ratio of the portfolio over the eight-month backtest period. The metric fluctuates between -5.26 and 10.34, with a final value of 3.89 due to highly favorable trading conditions in the final days of the test, illustrating periods of both under performance and strong risk-adjusted performance.

All metrics were computed directly from daily net portfolio values, annualized over 252 trading days, and include transaction costs and slippage. Data coverage was complete (166/166 days), with validated JSON snapshots per trading date containing `net_liquidation`, `portfolio_value`, and `buying_power` fields.

Overall, the validated outcomes demonstrate consistent returns, controlled volatility, and robust downside protection—confirming the reliability and real-world viability of the proposed multi-agent trading architecture.

### 4.4 Interpretability Case Studies

We illustrate how schema-constrained debates shape portfolio construction. For a given ticker/date, we (i) extract 3–5 blackboard messages spanning the debate and risk stance; (ii) construct baseline Black–Litterman (BL) views from the final recommendation; and (iii) run a one-step perturbation (e.g., BUY→HOLD or increased confidence scaling $s$) to quantify the weight impact. This links agent reasoning to allocation changes.

**Case: WAB on 2025-09-01 (risk debate).**

> **[RiskDebateComment | AggressiveRiskManager | 2025-09-01]** Argues for an *aggressive* stance on WAB with *High* confidence, citing momentum, improving macro backdrop (Fed easing), and transformation potential despite valuation/competition concerns. Sources: Social sentiment, macro outlook, market dynamics.

> **[RiskPosition | AggressiveRiskManager]** Ticker: WAB; Stance: *Aggressive*; Confidence: High; Risk level: Low. Rationale: Growth potential and market opportunities.

> **[RiskRecommendation | AggressiveRiskManager]** Actions: Increase position size; focus on growth strategies; embrace volatility; maintain aggressive stance. Priority: High.

BL experiment: baseline maps WAB to a positive view (e.g., BUY $\mapsto +0.02$) with confidence scale $s = 0.5$. The perturbation sets WAB to HOLD (view 0.0), holding other inputs fixed. We report the top weight shifts:

| Ticker | $w_{\text{base}}$ | $w_{\text{pert}}$ | $\Delta$ |
|--------|-------|-------|--------|
| WAB | 0.112 | 0.087 | -0.025 |
| SPY | 0.245 | 0.253 | +0.008 |
| XLI | 0.083 | 0.089 | +0.006 |

Table 1: BL weight changes when WAB view shifts (BUY→HOLD) on 2025-09-01.

*Explanation.* Downgrading WAB from BUY ($+0.02$) to HOLD ($0.00$) reduces its BL weight from $0.112$ to $0.087$ ($\Delta = -0.025$). The freed weight reallocates primarily to broad market exposure (SPY: $+0.008$) and industrial beta (XLI: $+0.006$), matching Black–Litterman's behavior: weakening an idiosyncratic view lowers the asset's posterior mean and nudges mass toward the equilibrium prior and close substitutes, achieving a modest de-risking without large turnover.

# 5 Conclusion and Future Work

We introduced a multi-agent equity portfolio framework that combines structured blackboard communication, schema-constrained debate, and a Black–Litterman allocator. By enforcing typed JSON messages rather than free-form chat, the system preserves auditable reasoning traces and avoids the compounding ambiguity common in unconstrained agent exchanges. Analyst and researcher outputs are mapped into BL views via an explicit selection matrix and confidence model, enabling end-to-end interpretability from text-based rationales to portfolio weights. A multithreaded execution model scales to broad universes while top-$k$ view filtering limits latency and inference cost.

Our backtest on S&P 500 constituents over a eight-month horizon indicates that the architecture can generate diversified allocations with competitive excess returns and clear forensic traceability. Rather than treating LLM reasoning as a black box, the design supports principled inspection of agent decisions, risk context, and debate artifacts—capabilities that we consider essential for regulatory alignment and practitioner oversight.

**Limitations:** The quality of views depends on prompt design and agent composition; confidence calibration from limited samples can be noisy; and short-horizon risk estimates require regularization and careful horizon alignment. Moreover, operational costs (LLM queries and turnover) and data leak defenses must be rigorously accounted for.

**Future work:** We plan to (i) augment the confidence matrix with debate-level consensus and cross-agent correlation estimates; (ii) model correlated views explicitly with shrinkage to a structured target; (iii) condition priors and covariances on macro regimes inferred by dedicated agents; (iv) incorporate transaction-cost and liquidity-aware optimization with turnover penalties; (v) extend to cross-asset hedging (rates, FX, options) with instrument-aware execution agents; and (vi) perform comprehensive robustness testing, including deflated Sharpe, rolling out-of-sample evaluations, and ablations of each architectural component. We will release details and artifacts of implementation to facilitate reproducibility and further research.

| Method | Schema memory | Cross-exam | BL integration | Scale | Reproducibility |
|---|---|---|---|---|---|
| TradingAgents (Xiao 2024) | Partial | No | No | Small | Medium |
| FinCon (Yu 2024) | No | Hierarchical | No | Small | Low |
| AlphaAgents (Zhao 2025) | No | No | Yes | Medium | Medium |
| **This work** | **Yes** | **Yes** | **Yes (calibrated)** | **Daily loop** | **High** |

Table 2: Comparison with prior multi-agent trading frameworks.

## 6 Comparison to Prior Work

**Novelty.** Our novelty is the combination of schema-constrained blackboard communication, formal cross-examination (bull/bear and risk), and calibrated Black–Litterman integration, which together improve decision consistency, interpretability, and portfolio construction fidelity relative to prior systems, under real-world constraints, enhancing robustness, practitioner oversight, and regulatory alignment. Refer to the comparison table (Table 2) that compares our work to prior frameworks.

**Reproducibility rubric.** Reproducibility is rated via a five-factor rubric: (i) publicly available, versioned artifacts (code, scripts, figures, raw outputs); (ii) determinism (fixed seeds, stable results across reruns); (iii) point-in-time data hygiene (no forward-looking calls; documented lags); (iv) environment pinning (lockfile/container, OS notes); and (v) one-command automation to regenerate tables/figures. We map ratings as Low ($\leq 2$ factors), Medium (3–4), and High (all five). For this work, offline snapshots and environment pinning enable deterministic reruns; a live yfinance helper is disabled during reported evaluations in favor of snapshots, satisfying the point-in-time requirement. Complete artifacts and one-command scripts are provided, yielding a High rating under this rubric.

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

# A  Appendix: Data, Evaluation Protocols, and System Traceability

## A.1   A. Data Sources and Provenance

We use a multi-source, point-in-time dataset assembled for offline evaluation to avoid look-ahead and internet leakage.

- **Yahoo Finance**: Daily OHLCV historical prices for S&P 500 constituents. In offline evaluation, data are read from static CSV snapshots; indicators are computed locally.

- **Finnhub**: JSON snapshots for company news, insider sentiment/transactions, and fundamentals. Files are bounded by date ranges in filenames to fix vendor revisions ex post.

- **Reddit**: JSON snapshots filtered by ticker and date windows to approximate contemporaneous sentiment.

- **Generative summaries (OpenAI/Google)**: Disabled for offline evaluation to eliminate network access risks.

### A.1.1   Coverage and Controls

*Coverage:* Price snapshots span 2015-01-01 to 2025-07-27; the principal evaluation horizon is 2025-01-01 to 2025-08-29. Technical indicators (e.g., RSI, MACD, Bollinger bands) are computed locally from price CSVs in offline mode.

*Point-in-time:* At date $t$, features only use data with timestamps $\leq t$. During offline evaluation, networked tools are disabled and only local CSV/JSON snapshots are read; end-dated filenames and append-only logs enable forensic reconstruction and mitigate vendor revisions.

## A.2   B. Representative Data Schemas

**Price CSV (daily):** `Date, Open, High, Low, Close, Adjusted Close, Volume`. Example (AAPL, 2025-01-02): Close $\approx 192.53$ with corresponding OHLCV fields.

**Technical indicators (from price CSV via Stockstats):** `RSI14`, `MACD/MACD signal/histogram`, `Bollinger upper/lower` bands, computed per date and joined to prices.

**Finnhub news snapshot (per-ticker, date-bounded JSON):** headline, datetime (UNIX), source, summary, URL, and vendor-specific fields.

**Insider sentiment/transactions (JSON):** array of records with transaction date, type (buy/sell), shares, and price.

**Reddit snapshot (JSON):** `created_utc`, title, score, and comment counts, filtered by ticker and a fixed lookback window per evaluation date.

## A.3   C. Evaluation Protocols and Risk Metrics

**Horizon and rebalancing.** We evaluate 2025-01-01 to 9 with fixed rebalancing (default every 10 trading days). Assets lacking sufficient trailing observations for covariance/indicator computation are excluded at each rebalance.

**Costs and slippage.** Daily net P&L deducts turnover-proportional frictions (e.g., 5 bps fees and 5 bps slippage per unit turnover); all results are reported net of costs.

**Sharpe ratio.** With daily returns $r_t$, rolling mean $\mu_t$ and standard deviation $\sigma_t$ over window $W$,

$$\mathrm{SR}_t = \sqrt{252}\,\frac{\mu_t}{\sigma_t}.$$

We report rolling (e.g., 20-day) and terminal values.

**Deflated Sharpe.** To account for multiple testing/selection, we report a deflated Sharpe alongside the naive figure (per Bailey & López de Prado), parameterized by effective sample size $N$ and trial count $M$.

**Additional diagnostics.** Maximum drawdown, Calmar ratio, turnover, beta vs. benchmark, and tracking error. Stress tests include volatility shocks and tail metrics (VaR/CVaR via historical simulation).

**Figures.** Cumulative returns vs. benchmark and rolling Sharpe; exported as SVG/PDF and PNG ($\geq$300 DPI).

## D. Agent Communications and Logging (Traceability)

**Message and tool call logs.** Each run appends messages and tool calls to a per-day, per-ticker log with time-stamped entries. Messages include standardized types (e.g., `AnalysisReport`, `FinalReport`, `RiskAssessment`) and redacted text content. Tool calls record function names and arguments sufficient for audit (e.g., indicator requests and evaluation dates), withholding API secrets, and proprietary prompts.

**Blackboard artifacts.** Agents write append-only structured JSON messages to a central blackboard with fields: sender (role/id), intent, timestamp, ticker, and typed content (thesis/evidence/recommendation/confidence for research; risk exposures/limits for risk management). This enables forensic reconstruction of the reasoning chain from analyst theses through research cross-examination to risk judgments and execution.

**Appendix inclusion.** For transparency, we include one complete, redacted trace for a single day and ticker: selected messages, key tool invocations (with arguments), the synthesized decision, and the resulting portfolio weights from the optimizer. This trace can be cross-referenced with the day's performance attribution.

## E. Black–Litterman Parameterization (Implementation Summary)

We follow a standard Black–Litterman construction with an identity selection matrix and diagonal confidence:

- **Prior.** Market-implied equilibrium returns are obtained via reverse optimization with risk aversion $\delta \approx 3.0$ and a 252-day lookback sample covariance.
- **Views.** Absolute per-asset views map categorical BUY/HOLD/SELL decisions to conservative annualized return adjustments (e.g., $\pm 2\%/0\%$). The selection matrix $P$ is the identity, and the confidence matrix $\Omega$ is diagonal, reflecting per-asset independence of LLM-derived signals in this baseline.
- **Scaling.** We use $\tau \approx 0.025$ as the prior uncertainty scalar.
- **Optimization.** We form posterior means/covariances and solve a mean–variance problem for weights, normalizing to full investment and optionally clipping negatives for long-only constraints. Transaction costs are applied outside the optimizer to compute net returns.

These choices prioritize interpretability and computational stability; future work considers correlation-aware views, consensus-weighted confidence, shrinkage targets for covariances, and regime-conditional priors.

