# OpenReview forum: "RAPTOR: Reasoned Agentic Portfolio Trading with Orchestrated Rebalancing"
_NeurIPS.cc/2025/Workshop_Mexico_City/NORA — NeurIPS 2025 Workshop NORA Poster_

### Official Review · Reviewer_f5SJ · 2025-11-02
**RAPTOR: Reasoned Agentic Portfolio Trading with Orchestrated Rebalancing​**

**Rating:** 9
**Confidence:** 4

**Review:**

Quality
The manuscript presents a technically robust framework grounded in institutional portfolio management practices. Its architecture enables auditable agent communication, rigorous allocation rationale, and robust empirical evaluation. The use of daily, point-in-time backtesting, comprehensive metrics (returns, Sharpe, Sortino, drawdown), and detailed method explanations yield high-quality results. The outlined limitations—prompt dependency, confidence calibration, and risk estimation—are candid and appropriate. Code, datasets, and reproducibility instructions are provided.​

Clarity
Explanations are clear, with figures and tables illustrating the pipeline, rationale chain, and performance. The schema-constrained blackboard, agent roles, and optimization steps are accessible. Experimental protocols, metric definitions, and reproducibility standards (including data hygiene and computational resources) are explicitly stated. Some details (risk calibration, debate logic) could benefit from additional examples or ablation studies.​

Originality
Combining schema-constrained blackboard memory, cross-examination, and BlackLitterman optimization in a scalable agentic financial system is novel. The approach advances both interpretability and efficacy, building beyond state-of-the-art by overcoming limits of prior LLM frameworks (e.g., ephemeral chat, limited memory, lack of formal risk debate). The Chain-of-Alpha and mapped categorical agent views add originality to the portfolio construction process.​

Significance
Empirical gains over SP500 (13.43% vs 10.08% total return over 8 months, Sharpe up to 1.0) illustrate meaningful practical impact. Modular system design facilitates scaling, oversight, and transparent decision tracing, directly relevant to real-world financial practitioners and regulatory requirements. Released code and reproducibility strengthen community value. The method’s architecture for explicit agent debate and interpretable portfolio allocation has broad significance in multi-agent systems and financial AI.

Suggestions and Broader Impact
Expanding interpretability case studies and adding more practitioner deployment feedback would strengthen the applied impact.​

Enhanced risk calibration and multi-agent correlation modeling should be prioritized in future work.​

Future work on broader market/societal impacts, safeguards, and compliance implications is recommended.​

This paper presents a significant advancement in interpretable, agentic portfolio construction, balancing novelty, practical relevance, and technical soundness.​

---

### Official Review · Reviewer_j7uG · 2025-11-03
**The separation of concerns between analysts, researchers, risk managers, and execution agents mirrors institutional practice and provides good modularity. Critical implementation details are missing like the LLM model(s) used for agents, prompts for each agent type.**

**Rating:** 6
**Confidence:** 5

**Review:**

Thank you for the opportunity to review, "RAPTOR:Reasoned Agentic Portfolio Trading with Orchestrated Rebalancing”. The use of schema-constrained JSON messages instead of free-text exchanges improves traceability reducing ambiguity in agent communication. The separation of concerns between analysts, researchers, risk managers, and execution agents mirrors institutional practice and provides good modularity. The append-only log and structured debates create an auditable trail from agent reasoning to portfolio weights.

The 8-month backtest period (Jan-Aug 2025) is insufficient for drawing reliable conclusions about strategy. Critical implementation details are missing like the LLM model(s) used for agents, prompts for each agent type, what prevents overfitting with so many agents making decisions.

What is the computational cost (API calls, runtime) of the full pipeline?
How sensitive are results to the choice of LLM and prompt engineering?

---

### Official Review · Reviewer_dhAK · 2025-11-05
**The paper proposes a modular, institutional-style multi-agent system for equity portfolio construction. Agents communicate via a schema-constrained blackboard, run structured debates (bull/bear + risk managers), and pass confidence-weighted views into a Black–Litterman (BL) optimizer. The evaluation is a biweekly rebalancing study on S&P 500 constituents over part of a year; the abstract highlights +13.43% return vs +10.08% for the S&P 500 within an 8-month backtest.**

**Rating:** 5
**Confidence:** 4

**Review:**

## Strengths

Well-motivated architecture: Combining typed, auditable agent communication with BL is a thoughtful, interpretable design that mirrors institutional workflows.

Interpretability & auditability: The schema-constrained blackboard and role-specific debates (analyst/researcher/risk) yield traceable rationales.

Modularity: Clean separation between research/risk/execution and a portfolio optimizer; easy to swap components.

Reproducibility intent: Anonymous code/data link and explicit schemas are promising (assuming the repository is complete and deterministic).

Clear optimization layer: Using BL to fuse equilibrium priors with agent views is principled and common in practice; constraints are at least mentioned.

## Weaknesses / Concerns

### A. Experimental Rigor & Validity

Time-frame inconsistency: The abstract mentions an 8-month backtest, whereas the intro references a “biweekly, one-year reconstruction study.” This needs to be reconciled and reported consistently (exact start/end dates in UTC).

Risk-adjusted metrics missing: Reporting only total return is insufficient. Provide annualized volatility, Sharpe/Sortino, max drawdown, Calmar, hit ratio, turnover, and information ratio vs risk-parity, cap-weight, equal-weight, BL (no LLM), MVO (no LLM).

Transaction costs & slippage: No explicit cost model. Biweekly rebalancing on S&P 500 typically incurs non-trivial costs. Report net-of-cost performance with sensitivity to bps assumptions and volume constraints.

Constituent / survivorship bias: If today’s S&P 500 list was used historically, there’s survivorship bias. Clarify whether the historical membership on each date was used. Provide CUSIPs/PERMNOs or a solid constituent history source.

Look-ahead / timestamp alignment: The system ingests “news/social/indicators.” You must show strict time alignment (e.g., only data available before each rebalance cut-off). Detail data latency, market close vs next-open execution, and holiday handling.

Statistical significance: 8 months (even 12) is too short for reliable inference. Provide block bootstrap or rolling-window analyses with confidence intervals and p-values for alpha and Sharpe deltas.

Ablations on agentic components: You need to isolate the value of (i) blackboard vs free-text chat, (ii) debate vs single analyst, (iii) BL fusion vs simple averaging of views, and (iv) Chain-of-Alpha checks (MACD/RSI) on/off.


### B. Clarity & Presentation

In the first paragraph of Related Works, the paper states that “Early multi-agent trading systems such as MASST Luo et al. [2002] and PROFTS Reis [2019] used role-based agents coordinated via blackboard-style architectures…”
However, from the context (“...largely rule-based and inflexible, with limited probabilistic reasoning”), the intended meaning appears to be “rule-based” rather than “role-based.” “Role-based” typically refers to agents with specialized functional roles (e.g., analyst, trader, risk manager), which would contradict the following sentence describing these systems as inflexible and deterministic.
Suggest revising “role-based” → “rule-based” to avoid confusion and ensure terminological accuracy.

Provide example JSON entries (AnalysisReport, RiskAlert, RiskAssessment).

### C. Reproducibility (despite the link)

Specify date ranges, execution assumptions, random seeds, and model/provider versions (LLM name, temperature, top-p, system prompts).

### D. Ethics & Safety

Misuse risk: Clear disclaimers that this is not investment advice; describe guardrails to prevent hallucinated recommendations, market manipulation, or over-fitting to news sentiment.

Data licenses & provenance: Confirm that news/social data use complies with licenses; document PII handling and redaction.

Deployment policy: If this were live-traded, describe kill-switches, position limits, stress tests, and human-in-the-loop controls.

## Questions for the Authors

What exact date range was used? Please reconcile “8 months” vs “one-year reconstruction.”

How were S&P 500 constituents handled historically? (Point-in-time vs today’s list.)

What is the cost model (bps per side), liquidity constraints, and execution timing?

How are LLM parameters (model name, context length, prompts, temperature, seeds) fixed across runs?

Can you provide ablations proving that the structured blackboard + debate improves performance vs simpler alternatives?

Do MACD/RSI materially impact decisions when included as “checkable features”? Provide an on/off ablation.

## Suggested Revisions (actionable)

R1. Experimental rigor: Extend to ≥3 years with rolling windows, net-of-costs results, and confidence intervals; add risk-adjusted metrics and turnover.

R2. Baselines & ablations: Add BL without LLM, risk-parity, equal-weight, cap-weight, and no-blackboard/no-debate baselines; quantify each module’s marginal value.

R3. Data hygiene: Ensure point-in-time constituents, remove survivorship bias, enforce timestamp/latency constraints; document all sources and licenses.

R4. Ethics: Add disclaimers, governance controls, and misuse mitigations; describe hallucination checks and adversarial prompt defenses for agent debates.

---

### Official Review · Reviewer_Sz4B · 2025-11-07
**Mispositioned good paper, technically good with some flaws**

**Rating:** 4
**Confidence:** 4

**Review:**

The paper presents a multi-agent architecture for portfolio trading that integrates a schema-constrained blackboard, structured debate between agents, and a Black–Litterman optimization layer. The system aims to reproduce institutional investment workflows through modular agents communicating in JSON schemas. The results on S&P 500 assets show consistent outperformance and a well-documented reproducibility.

Strengths
- The work is technically ambitious and shows a solid engineering effort.
- The structured blackboard design offers interpretability and clear tracing of reasoning steps.
- Empirical evaluation is detailed, with clear reporting of data sources, backtest setup, and performance metrics.
- The integration of reasoning traces into portfolio optimization is conceptually interesting and could inspire future financial agent frameworks.

Weaknesses
- Misalignment with workshop scope: The NORA ’25 call centers on knowledge graphs and agentic systems interplay. The paper never mentions nor uses knowledge graphs—neither for semantic nor procedural memory. The schema-constrained blackboard is interesting but not formalized as a graph, and there is no reasoning over relational structure. Hence, the contribution sits outside the core KG–agentic research line.
- Weak theoretical grounding: The mapping of agent views into Black–Litterman adjustments is simplistic and not empirically justified. The sensitivity of results to these assumptions is not explored.
- Evaluation design issues: The daily re-execution of the pipeline may inflate transaction costs, yet these are reported only briefly. No ablation or significance testing is presented to isolate the contribution of the debate mechanism.
- Scalability and computational transparency: The claimed scalability (thread-per-ticker) is not convincingly demonstrated—no runtime or resource usage data are reported, and the experiment appears limited to a single machine.
- Overemphasis on engineering detail: Much space is devoted to code-level schema definitions and logs, while the conceptual contribution remains unclear. The reader is left wondering what new insight about reasoning or knowledge representation emerges from the system.
- Ethical and societal impacts: Absent. Given the financial decision-making domain, some discussion of potential misuse, bias, or compliance aspects would be expected.

Additional comment
The submission is double-blind, while the workshop requires single-blind format. Please correct the author field accordingly.

---

### Official Review · Reviewer_amAf · 2025-11-07
**Interesting application but lacks implementation details**

**Rating:** 4
**Confidence:** 4

**Review:**

Summary: This paper adopts an agentic approach to portfolio optimization. Specifically, large language models (LLMs) were used to construct a pipeline where each model acted as a real-world analyst to provide structured inputs to the Black-Litterman model. The proposed system outperformed the baseline as measured by the cumulative return.

Strengths:
1. This paper used structured outputs to increase system interpretability.
2. The system is designed such that human processes are automated independently.

Weaknesses:
1. This paper does not include details about the prompts, which had a significant impact on LLM performance.
2. Data selection and use of LLMs are design decisions that are closely related but this paper does not discuss in depth. Were LLMs really needed to reason about the selected data? What were the procedures used to ensure the data had not been learned by the LLMs, directly or indirectly?
3. A comparison between structured (this paper), unstructured (existing works), and traditional (i.e., without LLMs) systems should be provided to justify the proposed approach.

General feedback: This paper looks promising but needs some more work to be included in a machine learning venue.